# Isoform-Specific Roles of ERK1 and ERK2 in Arteriogenesis

**DOI:** 10.3390/cells9010038

**Published:** 2019-12-21

**Authors:** Nicolas Ricard, Jiasheng Zhang, Zhen W. Zhuang, Michael Simons

**Affiliations:** 1Yale Cardiovascular Research Center, Department of Internal Medicine, Yale University School of Medicine, New Haven, CT 06511, USA; nic.ricard@gmail.com (N.R.); jiasheng.zhang@yale.edu (J.Z.); zhen.zhuang@yale.edu (Z.W.Z.); 2Department of Cell Biology, Yale University School of Medicine, New Haven, CT 06511, USA

**Keywords:** angiogenesis, arteriogenesis, ERK, VEGF, endothelial cells, inflammation, macrophages

## Abstract

Despite the clinical importance of arteriogenesis, this biological process is poorly understood. ERK1 and ERK2 are key components of a major intracellular signaling pathway activated by vascular endothelial growth (VEGF) and FGF2, growth factors critical to arteriogenesis. To investigate the specific role of each ERK isoform in arteriogenesis, we used mice with a global *Erk1* knockout as well as *Erk1* and *Erk2* floxed mice to delete *Erk1* or *Erk2* in endothelial cells, macrophages, and smooth muscle cells. We found that ERK1 controls macrophage infiltration following an ischemic event. Loss of ERK1 in endothelial cells and macrophages induced an excessive macrophage infiltration leading to an increased but poorly functional arteriogenesis. Loss of ERK2 in endothelial cells leads to a decreased arteriogenesis due to decreased endothelial cell proliferation and a reduced eNOS expression. These findings show for the first time that isoform-specific roles of ERK1 and ERK2 in the control of arteriogenesis.

## 1. Introduction

Blood vessel development and growth encompasses three distinct biological processes—vasculogenesis, angiogenesis, and arteriogenesis [1]. The term vasculogenesis denotes the formation of the primitive vascular plexus from progenitor cells in embryo and this mode of blood vessel formation is limited to embryonic development. Angiogenesis encompasses the process of new capillary formation from pre-existing capillary beds that involves proliferation, sprouting and, migration of endothelial cells. Finally, arteriogenesis refers to the growth of new arteries and arterioles either de novo or from pre-existing arterial collaterals [2]. Arteriogenesis, the process of new arterial vasculature growth, is critical to the restoration of tissue perfusion following the development of a functionally significant decrease of arterial inflow. It is important to note that compromised arterial inflow results in two distinct events: distal tissue ischemia that leads to local angiogenesis (e.g., angiogenesis along the myocardial infarction border zone) and arteriogenesis that occurs in close proximity to the site of arterial trunk occlusion, a territory that is typically not ischemic [3,4].

While molecular controls of angiogenesis are well understood, events triggering and regulating arteriogenesis are still a matter of intense study and controversy. Vascular endothelial growth factor (VEGF) is the main factor driving angiogenesis in response to tissue hypoxia [5], yet VEGF is equally critical to arteriogenesis [6]. Indeed, disruption of VEGF signaling and, in particular, a reduction in VEGF-induced endothelial ERK1/2 signaling, has been shown to result in decreased arteriogenesis [7]. 

While angiogenesis involves simple proliferation and sprouting of capillary endothelial cells, arteriogenesis requires a coordinated response that involves multiple cell types that, in addition to endothelial and vascular smooth muscle cells, include a panoply of inflammatory cells including lymphocytes [8,9], natural killer cells [10], macrophages [11], and mast cells [12]. The presence of inflammatory cells (and a local inflammatory response at an arteriogenic site) is critical as these cells serve as the major source of VEGF in the absence of tissue ischemia [13,14,15,16,17]. 

Endothelial response to VEGF stimulation encompasses a complex series of events that include activation of various intracellular signaling cascades [18]. Of these, activation of ERK1/2 has been directly linked to the extent of arteriogenesis [7]. Remarkably, little is known about the individual contribution of ERK isoforms to this process. Global deletion of *Erk2* is embryonic lethal [19] whereas a global deletion of *Erk1* has no apparent vascular phenotype [20]. Furthermore, endothelial-specific deletion of *Erk2* on the *Erk1* global knockout background is lethal early on in embryonic development due to impaired vascular development [21]. Another key role played by the two ERKs in the adult endothelium is the regulation of vascular normalcy and integrity [22].

In this paper, we focused on the role of ERK1/2 isoforms in adult arteriogenesis. Induction of acute hindlimb ischemia resulted in excessive but poorly functional arteriogenesis in mice with a global deletion of *Erk1* whereas mice with endothelial-specific deletion of *Erk2* exhibited a decreased arteriogenesis. Since arteriogenesis involves a number of cell types, we generated a floxed *Erk1* mouse line and conditionally deleted the gene in macrophages, endothelial, and smooth muscle cells. While endothelial or macrophage deletions of *Erk1* failed to recapitulate the phenotype of the *Erk1^−/−^* mice, combined deletion of *Erk1* in endothelial cells and macrophages came close to the phenotype in global *Erk1* null mice. Altogether, these results show that endothelial and macrophage *Erk1* is critical to endothelial/macrophage crosstalk and effective adult arteriogenesis.

## 2. Methods

### 2.1. Mice

*Mapk3^−/−^* mice (denominated *Erk1^−/−^*), *Mapk1*^tm1Gela^/J mice (denominated *Erk2*^Fl/Fl^), and B6.129P2-Lyz2tm1(cre)Ifo/J mice (denominated LysMCre) were purchased from the Jackson Laboratory. *Cdh5CreER^T2^* mice were a generous gift from Ralf Adams. Myh11CreER^T2^ mice were a generous gift from Dan Greif. All mice, including the wild type (WT) mice, are *Mus musculus* on a pure C57Bl6 genetic background. *Erk1^Fl/Fl^* mice were realized by inserting 2 loxP sequences in introns between exons 2 and 3 and exons 8 and 9 of the Erk1 gene. Tamoxifen injections to induce deletion by the Cdh5Cre or Myc11Cre were done with 5 injections of 1.5 mg of tamoxifen on 5 consecutive days. Control mice received the same quantity of tamoxifen. For retinal angiogenesis, 100 µg of tamoxifen were administrated by IP injections starting at P1 to P4. BrdU was injected 2 h prior to euthanasia. Animals were housed and used in accordance with protocols and policies approved by the Yale Institutional Animal Care and Use Committee.

### 2.2. Endothelial Cells, Macrophages, and Aortic Smooth Muscle Cell Isolations and Quantitative PCR

Endothelial cells were isolated from mouse livers and lungs. Briefly, livers and lungs were collected and digested in a solution of collagenase and dispase (Roche/Sigma Aldrich, St Louis, MO, USA). The suspensions were then washed and filtered. Endothelial cells were isolated using magnetic beads anti-Rat IgG (Invitrogen, Camarillo, CA, USA) previously coated with rat anti-mouse CD31 antibody (BD). After extensive washing, cells were lysed and RNA was isolated using PicoPure RNa isolation kit (ThermoFisher, Waltham, MA, USA) or cultured.

Macrophages were isolated from the peritoneal cavity as previously described [23]. Macrophages were selected using magnetic beads anti-Rat IgG (Invitrogen) previously coated with rat anti-mouse F4/80 antibody (Invitrogen). After extensive washing, cells were lysed, and RNA was isolated using PicoPure RNa isolation kit (ThermoFisher).

Smooth muscle cells were isolated from the aorta. Aortas were collected and digested in 175 U/mL collagenase (Worthington), 1.25 U/mL elastase (Worthington, Lakewood, NJ, USA), and HBSS for 25 to 30 min at 37 °C. Adventitia layer was then pulled out. Media and endothelium were cut and digested in 175 U/mL collagenase and 2.5 U/mL elastase in HBSS for 1 h at 37 °C. Endothelial cells were bound toon beads previously coated with rat anti-mouse CD31 antibody were used and discarded. The remaining smooth muscle cells were lysed and RNA was isolated using PicoPure RNa isolation kit (ThermoFisher).

cDNAs were synthetized with iScript Reverse Transcription Supermix (Bio-Rad, Hercules, CA, USA) and qPCRs were performed using SsoAdvanced Universal SYBR Green Supermix (Bio-Rad).

### 2.3. shRNA Infection

shRNA targeting ERK1 and ERK2 (Sigma-Aldrich, St Louis, MO, USA) were encapsulated into lentivirus that were then. Lentivirus were produced in 293T cells using second generation lentiviral system (Invitrogen).

### 2.4. Hindlimb Ischemia Model

This was done as previously described by our lab [6]. Laser Doppler flow-imaging was carried out using a Moor Infrared Laser Doppler Imager (LDI; Moor Instruments Ltd., Wilmington, DE, USA) under ketamine and xylazine anesthesia.

### 2.5. Micro-CT Imaging

Microcomputed tomography (micro-CT) of the hindlimb vasculature was done by injecting 0.7 mL bismuth contrast solution in the descending aorta and the vasculature was imaged and quantified as previously described [6].

### 2.6. Western Blot

Cells were lysed in RIPA buffer (Boston BioProducts, Ashland, MA, USA). Proteins were titrated using Bio-Rad Protein Assay Dye Reagent (Bio-Rad). A total of 20 ng of proteins were loaded on a 4–12% acrylamide gel (Bio-Rad) and then transferred on a PVDF membrane (Millipore). Primary antibodies used were: F4/80 (Invitrogen), ERK (Cell Signaling, Danvers, MA, USA), and β-actin (Sigma-Aldrich).

### 2.7. Immunofluorescent Staining

Frozen sections were treated with ice cold acetone. Permeabilization was done in triton 0.1%. Primary antibodies were: F4/80 (Invitrogen) and IsolectinB4 (Invitrogen). Pictures were taken using an SP5 confocal microscope (Leica, Allendale, NJ, USA).

### 2.8. xCELLigence Real-Time Cell Analysis (RTCA)

Endothelial cell proliferation was measured by using an xCELLigence RTCA instrument (Roche Diagnostics) and E-plate 16 (a modified 16-well plate, Roche Diagnostics). The E-plate 16 was coated with 0.1% gelatin, loaded with 100 μL cell-free medium, and left in a tissue culture hood for 30 min to reach equilibrium. The E-plate 16 was placed into the RTCA instrument to measure the background impedance. Thereafter, 100 μL cell suspensions with fewer than 3500 cells were added into each well of the E-plate 16, which was then placed in a tissue culture incubator for 30 min to allow cells to settle down before being measured by the RTCA device. The impedance value of E-plate 16 was automatically monitored every 15 min. FGF2 was added at 100 ng/mL.

### 2.9. Endothelial Migration

HUVEC migration was measured in a wound-healing assay, which used ibidi Culture-Inserts (ibidi) to generate the wound. An ibidi Culture-Insert has dimensions of 9 mm × 9 mm× 5 mm (width × length × height) and is composed of two wells. One or two inserts were placed into one well of a six-well plate. After being coated with 0.1% gelatin, both wells of inserts were loaded with 100 μL cell suspension. FGF2 and VEGF were used at 100 ng/mL. 

### 2.10. Statistical Analyses

Statistical tests were performed using the software GraphPad Prism 8. 

## 3. Results

### 3.1. Erk1 KO Mice Exhibit Excessive but Poorly Functional Arteriogenesis

To assess the role of ERK1 in arteriogenesis, we ligated the common femoral artery (CFA) of *Erk1^−/−^* mice and assessed blood flow recovery over time using laser Doppler imaging while the anatomical extent of arteriogenesis was studied using micro-CT. While blood flow recovery was reduced in *Erk1^−/−^* mice compared to wild type (WT) controls (Figure 1A,B), the micro-CT-determined extent of arteriogenesis was dramatically increased (Figure 1C,D). Since macrophages are the critical source of VEGF in this model and since the extent of arteriogenesis generally correlates with the amount of VEGF-A present [24], we used immunocytochemistry to assess the extent of macrophage accumulation in the arteriogenic zone. Staining with an F4/80 antibody of the thigh muscles around the area of CFA, ligation was carried out three and seven days following hindlimb ischemia. There was a marked increase in macrophage tissue infiltration in *Erk1^−/−^* mice compared to WT mice (Figure 1E). To confirm these findings, we carried out Western blotting using total thigh muscle lysates. In agreement with immunocytochemical findings, we observed a massive increase in F4/80 signal (Figure 1F).

### 3.2. Erk1 Deletions in Endothelial Cells, Macrophages, or Smooth Muscle Cells Do Not Affect Arteriogenesis 

To identify the cell type(s) involved in the defective arteriogenic phenotype seen in *Erk1^−/−^* mice, we created mice with two intronic loxP sites in the *Erk1* (*Mapk3*) gene (hereafter denominated as *Erk1^Fl/Fl^* (Figure 2A)). Since endothelial cells are critical to arteriogenesis [6], these mice were crossed with a strain carrying an inducible Cre recombinase under the control of the VE-cadherin promotor (Cdh5CreER^T2^) [25] generating a Cdh5CreER^T2^;Erk1^f/f^ line (*Erk1^iEC−/−^*). Administration of tamoxifen to eight-week-old *Erk1^Fl/Fl^* mice resulted in a high efficiency deletion of endothelial *ERK1* (Figure 2B). One week after tamoxifen treatment, common femoral arteries (CFA) of *Erk1^iEC−/−^* and control mice were ligated. Surprisingly, laser-Doppler assessment of blood flow recovery in *Erk1^iEC−/−^* mice showed that it was similar to that of WT control mice (Figure 2C,D). We next turned our attention to macrophages. Similarly to Cdh5 CreER^T2^, LysM Cre [26] was very effective in deleting *Erk1* (Figure 2E). Blood flow recovery after CFA ligation was not impaired compared to WT mice after either *Erk1* gene deletion (Figure 2F,G). We next deleted the *Erk1* gene in smooth muscle cells (SMC) using the Myh11CreER^T2^ driver line [27]. *Erk1* gene was efficiently deleted in SMC (Figure 2H) and this deletion had no effect on the blood flow recovery after CFA (Figure 2I,J).

### 3.3. Erk1 Deletions in Endothelial Cells and Macrophages Leads to an Excessive but Poorly Functional Arteriogenesis

Finally, we bred mice with macrophage- (*Erk1*^MϕKO^) and endothelial (*Erk1*^iECKO^)-specific deletions to generate double knockout mice with ERK1 expression disrupted in both cell types (*Erk1*^iECKO/MϕKO^). Induction of hindlimb ischemia in these mice led to impaired blood flow recovery that was similar to that observed in the *Erk1* global null mice (Figure 3A,B) and the anatomical extent of arteriogenesis was increased (Figure 3C,D). We carried out Western blotting using total thigh muscle lysates and we observed a massive increase in F4/80 signal (Figure 3E). 

### 3.4. Erk2 Deletions in Endothelial, but Not Other Cell Types, Decreases Arteriogenesis

We next deleted endothelial *Erk2* (*Mapk1*) using the same Cdh5CreER^T2^ line. As in the case of *Erk1*, the administration of tamoxifen to eight-week-old *Erk2^Fl/Fl^* mice resulted in high efficiency deletion of the endothelial *Erk2* (Figure 4A). However, unlike the *Erk1^iEC−/−^* mice, the deletion of endothelial *Erk2* resulted in reduced flow recovery in *Erk2^iEC−/−^* compared to WT mice (Figure 4B,C). Surprisingly, there was no difference in the anatomical extent of arteriogenesis as determined by micro-CT imaging (Figure 4D,E). Endothelial nitric oxide synthase (eNOS) is a key enzyme producing the vasodilator NO and its activity is critical to arteriogenesis [28]. We found a decreased eNOS expression in *Erk2* KO endothelial cells compared to endothelial cells from WT mice (Figure 4F). We next focused on macrophages. LysM Cre was very effective in deleting *Erk2* in mice (Figure 4G). However, blood flow recovery after CFA ligation was not impaired compared to WT mice after *Erk2* gene deletion (Figure 4H,I). We next deleted the *Erk2* gene in smooth muscle cells using Myh11CreER^T2^ driver line (Figure 4J). *Erk2* deletion had no effect on the blood flow recovery (Figure 4K,L).

### 3.5. ERK Isoform Effect on Endothelial Cell Proliferation and Migration

To gain an insight into differences in ERK1- vs. ERk2-specific effects in the endothelium, we examined the effect of either isoform deletion on endothelial cell proliferation and migration. Administration of BrdU to P6 *Erk1^−/−^* mice showed no differences in the extent of endothelial cell proliferation in the retinal vasculature vs. WT controls (Figure 5A,B). In contrast, BrdU labeling in *Erk2^iECKO^* mice showed a ~50% reduction in endothelial proliferation. A combination of EC-specific *Erk2* and global *Erk1* knockouts (*Erk2^iECKO^*; *Erk1^−/−^* mice) did not result in a further decline in endothelial proliferation demonstrating that *Erk2* is the primary driver of this process. These findings were confirmed in vitro: *Erk2* but not *Erk1* knockdown resulted in decreased EC proliferation in an FGF2 growth assay (Figure 5C). In contrast, both ERKs were involved in EC migration response to VEGF-A or FGF2 stimulation in in vitro cell wounding assays (Figure 5D).

## 4. Discussion

The results of this study show that ERK isoforms have a differential effect on arteriogenesis. While a global *Erk1* knockout impaired blood flow recovery due to inefficient arteriogenesis, it took a combination of endothelial- and macrophage-specific knockout of this isoform to match the global deletion phenotype. The principal driver of response appeared to be a large increase in tissue macrophage levels that likely resulted in abnormally high VEGF levels and exuberant, albeit inefficient, arteriogenesis. In contrast, endothelial-specific deletion of the *Erk2* isoform resulted in reduced blood flow recovery even though the anatomical extent of arteriogenesis appeared normal. The culprit in this case was a dramatic reduction in endothelial eNOS expression that led to vasoconstriction. Neither ERK isoform deletion by itself in smooth muscle cells affected either blood flow recovery or arteriogenesis per se, while deletion of both genes resulted in a transient decrease of blood flow recovery.

Arteriogenesis is a process leading to the formation of arteries and arterioles. It can proceed either by remodeling of pre-existing collateral arteries or by expansion and arterialization of the capillary bed [2,5,29]. Arteriogenesis is distinct from angiogenesis, a process defined as sprouting and proliferation of the existing capillary bed [2]. Importantly, the two processes are regulated by distinctly different sets of factors. While hypoxia is the primary driver of angiogenesis, arteriogenesis is induced by a combination of shear stress and other mechanical factors [2,5,30]. At the same time, VEGF, and its subsequent induction of endothelial ERK activation, are crucial to both angiogenesis and arteriogenesis [31,32,33,34,35]. One important distinction, however, is the source of VEGF: while, in angiogenesis, VEGF is produced locally due to tissue ischemia, in arteriogenic settings macrophages are the key source of the growth factor [24].

While the importance of the VEGF/VEGFR2/ERK signaling cascade in both angiogenesis and arteriogenesis has been clearly recognized, how this signaling cascade promotes the two distinct means of vascular growth has been unclear. ERK activation is thought to be involved in the proliferation and migration of endothelial cells. Interestingly, our data indicate that endothelial proliferation is largely controlled by ERK2 while migration is the additive function of both isoforms. These results are in agreement with the study of Lefloch et al. who found that ERK2 controls cell proliferation in NIH 3T3 cells [36]. Other isoforms-specific effects of ERK signaling are regulation of macrophage accumulation by ERK1 and regulation of eNOS expression by ERK2.

Both global and a combination of macrophage- and/endothelial-specific *Erk1* knockouts led to markedly increased macrophage accumulation at the site of arteriogenesis after the common femoral artery ligation that was coupled with exuberant but ineffective arteriogenesis. The critical role macrophages play in arteriogenesis is well established. While the M2 subset (macrophages involved in wounds healing and vascular growth) have been described as a principle source of VEGF [37,38], other cell population, including blood-derived inflammatory cells, and mechanical factors can also contribute [13,39]. The observed increase in tissue macrophage levels in these mutant strains is likely derived from circulating monocytes [11] although a proliferation of resident M2 macrophages cannot be ruled out [39,40]. Endothelial cells play a crucial role in monocyte recruitment by increasing expression of the Notch ligand Dll1 [11], as has been observed in this study. Activation of Notch signaling in recruited monocytes polarizes them to an arteriogenic M2 phenotype [11]. Similarly, haploinsufficiency of *Phd2*, encoding the PHD2 oxygen sensor, leads to an expansion of tissue-resident M2-like macrophages, an increased release in arteriogenic factors, and an improved vascular reperfusion in the hindlimb ischemia model [37]. Here, we found that ERK1 controls endothelial-macrophage crosstalk, and that exacerbated macrophage infiltration increases arteriogenesis extent but decreases arteriogenesis functionality. However, a combination of the macrophage- and endothelial-specific *Erk1* knockout phenotype is not as severe as in the *Erk1* global knockout mice, suggesting that other cell types are also may be involved in the phenotype found in the *Erk1* global knockout mice.

In addition to the Dll1 signaling, endothelial cells also regulate macrophage infiltration via MAPK pathways. There are four distinct MAPK pathways: ERK1/2, ERK5, p38, and JNK. While we show that ERK1 controls macrophage recruitment, p38 MAPK pathway has also been shown to be involved in this process [41]. Indeed, a p38 downstream effector MAP-kinase-activated protein kinase 2 (MK2) induces MCP-1 expression in endothelial cells, which promotes monocyte chemoattraction. Thus, at least two different MAPK pathways are involved in the promotion of macrophage infiltration.

Effective blood flow recovery requires not only expansion of the arterial bed thereby ensuring adequate blood supply, but also effective organization and function of this newly formed vasculature. Interestingly, these processes appear to be differentially regulated. We and others have previously reported a dissociation between the extent of anatomical arteriogenesis and effective blood flow. Thus, a mouse strain with an endothelial loss of NF-kB signaling due to deletion of *Rela* showed increased and disorganized arteriogenesis and decreased tissue perfusion after CFA ligation [42]. This was driven by decreased expression of Dll4 that is NF-kB dependent. Delta-like 4 (Dll4) promotes arterial differentiation and restricts vessel branching by direct endothelial cell–cell signaling. Indeed, a similar phenotype was observed in adult *Dll4*^+/−^ mice. These animals also show reduced blood flow recovery after femoral artery occlusion despite exuberant arteriogenesis [43]. Finally, mice with an endothelial-specific deletion of HIF2α also display increased arteriogenesis abut impaired blood flow recovery in the same hindlimb model [44]. Our description of excessive arteriogenesis yet impaired perfusion in ERK1 null mice adds to this growing body of literature.

In contrast to *Erk1*, *Erk2* knockout in endothelial cells resulted in impaired blood flow recovery despite the normal anatomical extent of arteriogenesis. This is likely due to a decrease in endothelial proliferation combined with decreased expression of eNOS and a corresponding fall in NO production that is critical to the maintenance of arterial tone. Indeed, these observations match a similar decrease in blood flow recovery in endothelial eNOS knockout mice.

ERK1 and ERK2 isoforms share 84% of their amino-acid sequences. ERK1 is larger than ERK2 due to an extension of 17 amino-acids at its N-terminal and two amino-acids at its C-terminal. It has been long a matter of debate whether ERK1 and ERK2 have isoform-specific functions or are totally redundant [45]. ERK2 is expressed at higher levels than ERK1 in most mammalian tissues [46,47]. This difference in expression level may account for the difference in phenotype of the global knockout. Indeed, *Erk1^−/−^* mice are viable and fertile [20], whereas *Erk2^−/−^* mice die at an early stage in development [19]. Several studies support the functional redundancy of ERK1 and ERK2. Indeed, deletion of *Erk2* but not *Erk1* affected NIH 3T3 cell proliferation in vitro while overexpression of *Erk1* in *Erk2*-deficient NIH 3T3 cells rescued this proliferation defect [36]. ERK1 can also rescue the loss of ERK2 in vivo. Indeed, overexpression of *Erk1* in *Erk2^−/−^* mice, generating mice expressing only the ERK1 isoform, fully rescue the developmental defects associated with the loss of ERK2 [47]. We recently published that loss of endothelial *Erk2* in a global *Erk1^−/−^* background in adult mice is lethal, whereas loss of one of the two isoforms in the endothelium of adult mice has no vascular phenotype [22]. Interestingly, deletion of *Erk2* by a ubiquitously expressed Cre in adult *Erk1^−/−^* mice is lethal in less than three weeks due to multiple organ failure [48]. However, adult mice with only one allele of ERK regardless of the isoform survive [48]. These results suggest redundant roles between ERK1 and ERK2. On the other hand, other studies suggest isoform-specific functions (review in [45]). Notably, in a model of myocardial ischemia/reperfusion injury, myocardial infarction extent was found to be similar in *Erk1^−/−^* mice and WT mice [49]. However, mice lacking one *Erk2* allele (*Erk2*^+/−^), developed increased infarct areas compared with WT mice.

In summary, our data demonstrate specific roles of ERK isoforms in endothelial cells. While ERK1 controls macrophage infiltration following an ischemic event, ERK2 primarily controls endothelial cell proliferation and eNOS expression. Both isoforms are involved in regulation of migration.

## Figures and Tables

**Figure 1 cells-09-00038-f001:**
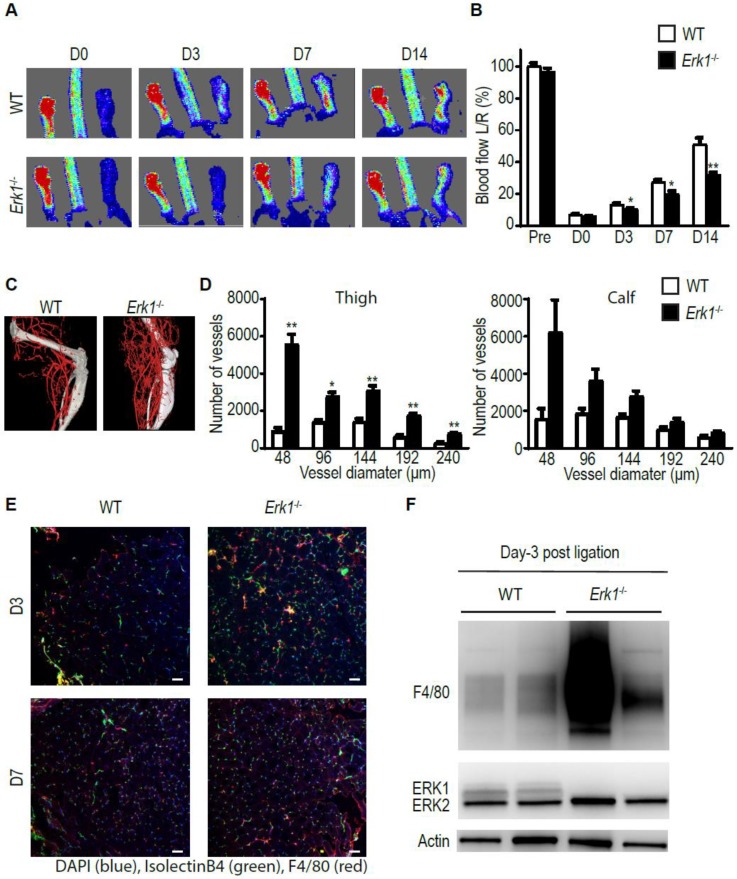
*Erk1^−/−^* mice exhibit excessive but dysfunctional arteriogenesis. (**A**,**B**) Blood flow recovery after ligation of the CFA in *Erk1^−/−^* mice assessed by laser-Doppler right after the surgery and 3, 7, and 14 days thereafter. Bar graph represents mean with SEM (*n* = 10 mice) * *p* < 0.05, ** *p* < 0.005 using two-way ANOVA followed by Sidak’s multiple comparison test. (**C**,**D**) Quantification of the hindlimb vasculature by micro-CT three weeks after femoral artery ligation. Bar graph represents mean with SEM (*n* = 4 mice) * *p* < 0.05, ** *p* < 0.005 using two-way ANOVA followed by Sidak’s multiple comparison test. (**E**,**F**) Macrophage infiltration in the thigh from *Erk1^−/−^* mice assessed by staining or Western blotting. Scale bar, 50 µm. Bar graph represents mean with SEM (*n* = 3 mice) * *p* < 0.05 using *t*-test.

**Figure 2 cells-09-00038-f002:**
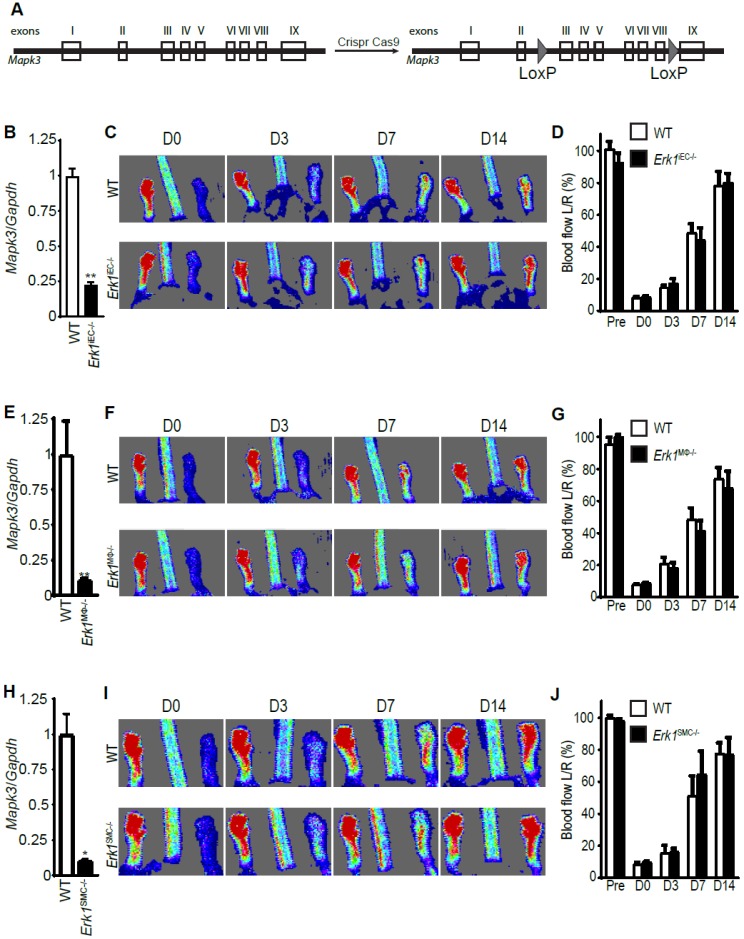
*Erk1* deletions in endothelial cells, macrophages, or smooth muscle cells do not affect arteriogenesis. (**A**) Generation of *Erk1* floxed mice by insertion of LoxP sites between exons 2 and 3 and exons 8 and 9. (**B**) Efficiency of *Erk1* deletion in endothelial cells was assessed by Q-PCR of endothelial cells isolated from mouse livers. Bar graph represents mean with SEM (*n* = 4 mice) ** *p* < 0.005 using Mann–Whitney test. (**C**,**D**) Blood flow recovery after ligation of the CFA in *Erk1*^iEC*−/−*^ mice assessed by laser Doppler right after the surgery and 3, 7, and 14 days after the surgery. Bar graph represents mean with SEM (*n* = 8 mice). (**E**) Efficiency of *Erk1* deletion in macrophages was assessed by Q-PCR of peritoneal macrophages. Bar graph represents mean with SEM (*n* = 4 mice) ** *p* < 0.005 using Mann–Whitney test. (**F**,**G**) Blood flow recovery after ligation of the CFA in *Erk1*^Mϕ*−/−*^ mice assessed by laser-Doppler right after the surgery and 3, 7, and 14 days thereafter. Bar graph represents mean with SEM (*n* = 6 mice). (**H**) Efficiency of *Erk1* deletion in smooth muscle cells was assessed by Q-PCR of smooth muscle cells isolated from the mouse aortas. Bar graph represents mean with SEM (*n* = 4 mice) * *p* < 0.05 using Mann–Whitney test. (**I**,**J**) Blood flow recovery after ligation of the CFA in *Erk1*^SMC*−/−*^ mice assessed by laser-Doppler right after the surgery and 3, 7, and 14 days after the surgery. Bar graph represents mean with SEM (*n* = 5 mice).

**Figure 3 cells-09-00038-f003:**
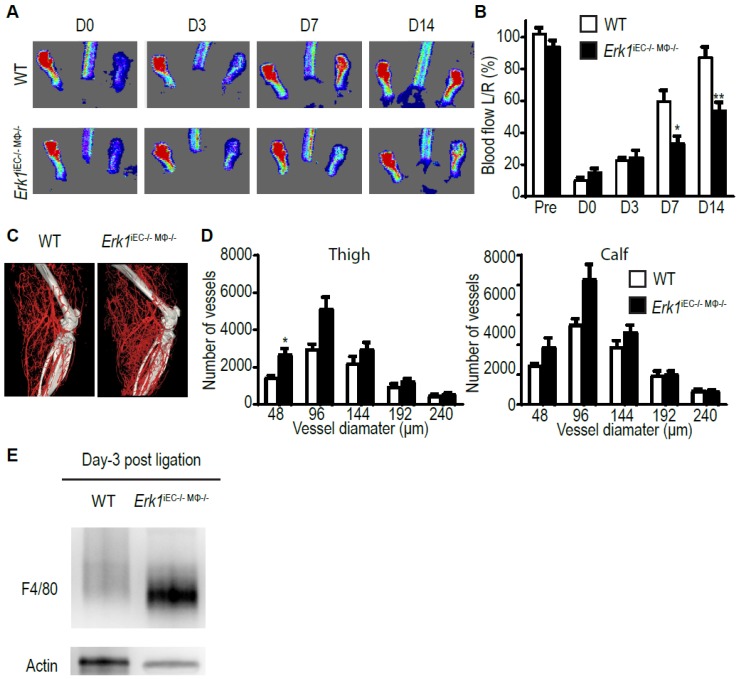
*Erk1* deletion in macrophages and endothelial cells leads to an excessive but poorly functional arteriogenesis. (**A**,**B**) Blood flow recovery after ligation of the femoral artery in *Erk1*^iEC*−/−*Mϕ*−/−*^ mice assessed by laser-Doppler right after the surgery and 3, 7, and 14 days thereafter. Bar graph represents mean with SEM (*n* = 6 mice) * *p* < 0.05, ** *p* < 0.005 using two-way ANOVA followed by Sidak’s multiple comparison test. (**C**,**D**) Quantification of the hindlimb vasculature by micro-CT three weeks after femoral artery ligation. Bar graph represents mean with SEM (*n* = 6 mice) * *p* < 0.05 using two-way ANOVA followed by Sidak’s multiple comparison test. (**E**) Macrophage infiltration in the thigh from *Erk1*^iEC*−/−*Mϕ*−/−*^ KO mice assessed by Western blotting. Bar graph represents mean with SEM (*n* = 3 mice) * *p* < 0.05 using *t*-test.

**Figure 4 cells-09-00038-f004:**
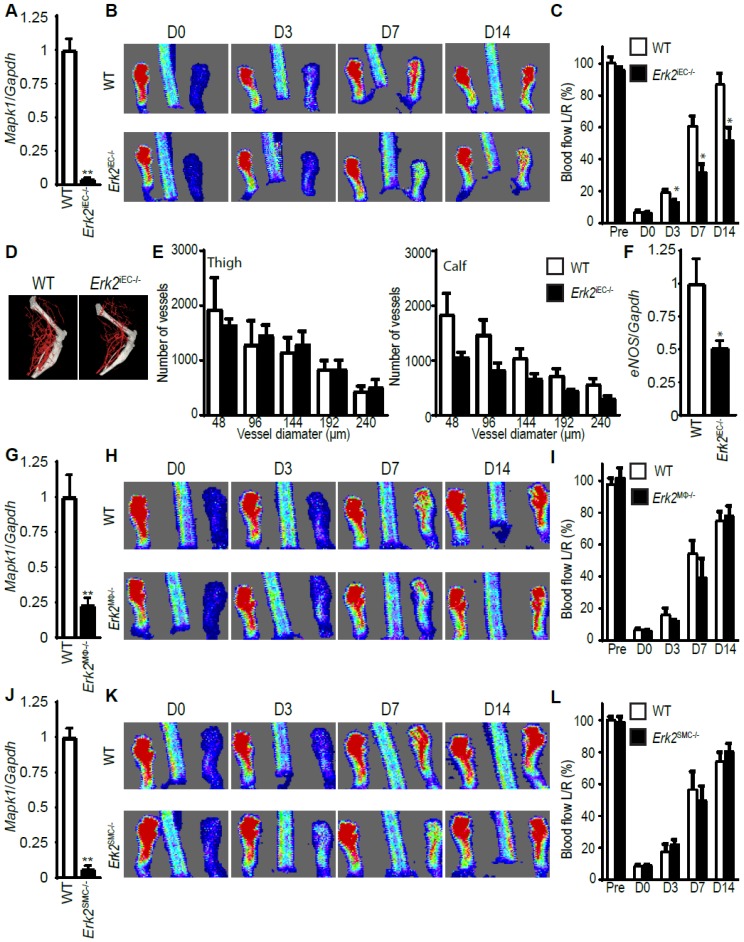
*Erk2* deletion in endothelial cells, but not in other cell types, decreases arteriogenesis. (**A**) Efficiency of *Erk2* deletion in endothelial cells was assessed by Q-PCR of endothelial cells isolated from mouse livers. Bar graph represents mean with SEM (*n* = 4 mice) ** *p* < 0.005 using Mann–Whitney test. (**B**,**C**) Blood flow recovery after ligation of the CFA in *Erk2*^iEC*−/−*^ mice assessed by laser-Doppler right after the surgery and 3, 7, and 14 days thereafter. Bar graph represents mean with SEM (*n* = 5 mice) * *p* < 0.05 using two-way ANOVA followed by Sidak’s multiple comparison test. (**D**,**E**) Quantification of the hindlimb vasculature by micro-CT three weeks after femoral artery ligation. Bar graph represents mean with SEM (*n* = 3 mice). (**F**) eNOS expression in endothelial cells isolated from livers from *Erk2*^iEC*−/−*^ mice. * *p* < 0.05 using Mann–Whitney test. (**G**) Efficiency of *Erk2* deletion in macrophages was assessed by Q-PCR on macrophages from the peritoneal cavity. Bar graph represents mean with SEM (*n* = 4 mice) ** *p* < 0.005 using Mann–Whitney test. (**H**,**I**) Blood flow recovery after ligation of the femoral artery in *Erk2*^MΦ*−/−*^ mice assessed by laser-Doppler right after the surgery and 3, 7, and 14 days thereafter. Bar graph represents mean with SEM (*n* = 4 mice). (**J**) Efficiency of *Erk2* deletion in smooth muscle cells was assessed by Q-PCR of smooth muscle cells isolated from mouse aortas. Bar graph represents mean with SEM (*n* = 4 mice) ** *p* < 0.005 using Mann–Whitney test. (**K**,**L**) Blood flow recovery after ligation of the femoral artery in *Erk2*^SMC*−/−*^ mice assessed by laser-Doppler right after the surgery and 3, 7, and 14 days thereafter. Bar graph represents mean with SEM (*n* = 6 mice).

**Figure 5 cells-09-00038-f005:**
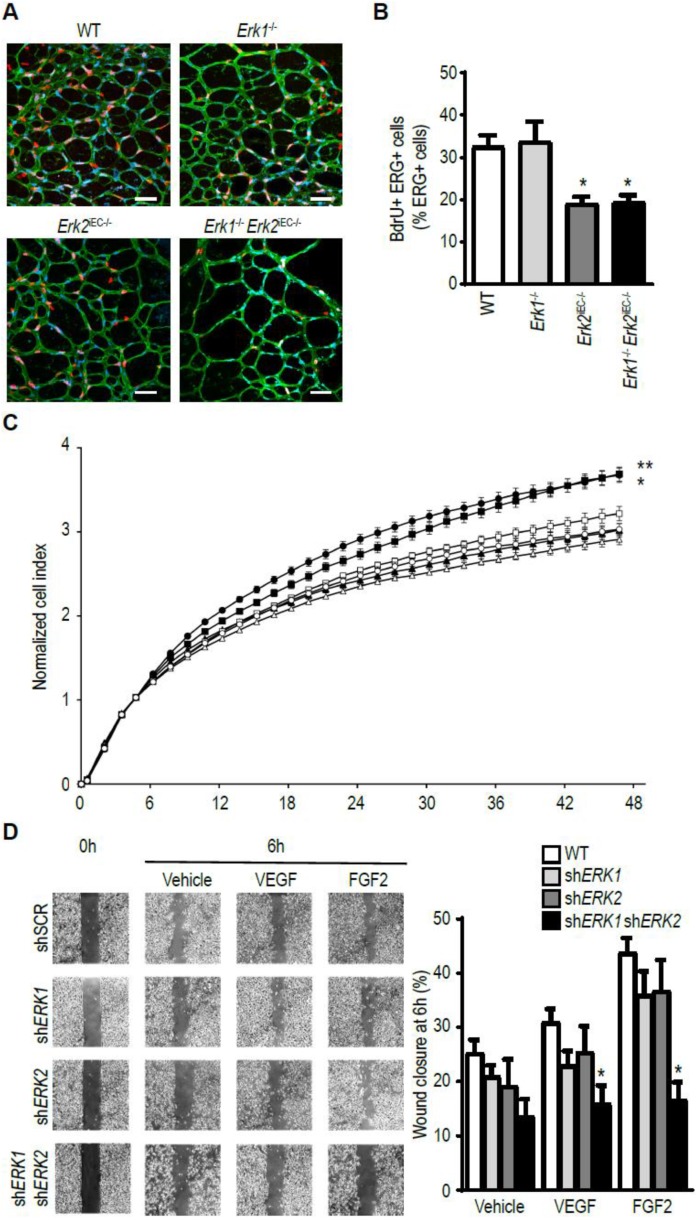
ERK isoform effect on endothelial cell proliferation and migration. (**A**,**B**) Assessment of endothelial cells proliferation in the retina of six-day-old pups by BrdU quantification. * *p* < 0.05 compared to WT using Kruskel–Wallis test with Dunn’s multiple comparison test. Scale bar, 50 µm. Bar graph represents mean with SEM (*n* = 5 mice). (**C**) Assessment of proliferation of pulmonary endothelial cells from WT mice (circle), *Erk1^−/−^* mice (square), or *Erk2*^iECKO^ mice (triangle). White symbols are cells treated with vehicle, and black symbols cells treated with FGF2. Bar graph represents mean with SEM (*n* = 6 wells) * *p* < 0.05, ** *p* < 0.005 compared to WT treated with vehicle using Kruskel–Wallis test with Dunn’s multiple comparison test. (**D**) Assessment of migration of HUVEC treated with shRNA scrambled, or against *ERK1*, or against *ERK2* and stimulated with (VEGFA_165_ or FGF2. Bar graph represents mean with SEM (*n* = 4 wells) * *p* < 0.05 compared to siSCR of each condition using Kruskel–Wallis test with Dunn’s multiple comparison test.

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
