# Peer review of "Isoform-Specific Roles of ERK1 and ERK2 in Arteriogenesis"

_cells, 2019, doi:10.3390/cells9010038_

Round 1

Reviewer 1 Report

Title: Isoform specific roles of Erk1 and Erk2 in arteriogenesis

Summary:

In the present study Ricard et al. investigated the isoform specific roles of Erk1 and Erk2 in Erk1 global knockout as well as in cell-type specific knockout mice after femoral artery ligation. They report that in global Erk1 deficient mice blood perfusion is decreased, although microCT displayed increased arteriogenesis. Endothelial cell, macrophage or vSMC-specific Erk1 knockout showed no difference compared to wildtype mice. Double EC/macrophage Erk1 knockout resembles the global knockout phenotype with increased F4/80 expression. On the contrary, Erk2 specific deletion in ECs, but not in macrophages or vSMCs, showed reduced blood perfusion, however microCT analysis did not display differences between EC-specific Erk2 knockout and wildtype mice. The authors explain these differences by enhanced recruitment of M2 macrophages in Erk1 knockout mice and with reduced eNOS expression in EC-specific Erk2 knockout mice.

General comments:

The findings reported in this manuscript are novel, interesting and of high interest to a broad readership. However, some important (control) experiments and information are missing to support conclusions drawn from the presented results.

Major comments:

All Figures: Did the wildtype control mice receive tamoxifen injections? In each experiment the number of animals used (n) is missing. Please state in each figure how many mice are used or how often the cell culture experiments are repeated. Figure 1: E/F Quantification of F4/80 stained macrophages in the immunohistochemisty and western blot analysis is missing. Moreover, to support the conclusion that these macrophages are M2 macrophages marker staining for M1 (CD11c/CD80) and M2 (CD206) should be performed. At least a VEGF staining should be performed. Figure 3: E: Quantification of western blot analysis is missing. Immunohistochemistry staining for F4/80 should be performed to allow comparison to global Erk1 knockout mice. Furthermore, eNOS expression in endothelial cells should be analysed to compare to EC-specific erk2 knockout effect. Figure 4: Immunohistochemistry staining for F4/80 and VEGF should be performed to allow comparison to global Erk1 and EC/macrophage-specific Erk1-/- knockout mice. Figure 5: If pulmonary ECs from the different knockout mice are available, I would suggest analysing eNOS expression in these cells in addition to LSECs.

Minor comments:

Introduction:

p1, line 26 ”is particular is” p1, line 26-31, citation? p1, line 45, ..signaling cascades VEGF.

Methods:

Method description for isolation of aortic vSMCs and pulmonary ECs is missing. siRNA transfection protocol is missing. There is a typo on page 13, line 289, “beqads”

Author Response

Summary:

In the present study Ricard et al. investigated the isoform specific roles of Erk1 and Erk2 in Erk1 global knockout as well as in cell-type specific knockout mice after femoral artery ligation. They report that in global Erk1 deficient mice blood perfusion is decreased, although microCT displayed increased arteriogenesis. Endothelial cell, macrophage or vSMC-specific Erk1 knockout showed no difference compared to wildtype mice. Double EC/macrophage Erk1 knockout resembles the global knockout phenotype with increased F4/80 expression. On the contrary, Erk2 specific deletion in ECs, but not in macrophages or vSMCs, showed reduced blood perfusion, however microCT analysis did not display differences between EC-specific Erk2 knockout and wildtype mice. The authors explain these differences by enhanced recruitment of M2 macrophages in Erk1 knockout mice and with reduced eNOS expression in EC-specific Erk2 knockout mice.

General comments:

The findings reported in this manuscript are novel, interesting and of high interest to a broad readership. However, some important (control) experiments and information are missing to support conclusions drawn from the presented results.

We would like to thank the reviewer for the useful insights.

Major comments:

All Figures: Did the wildtype control mice receive tamoxifen injections?

When tamoxifen was injected (Cdh5-Cre and Myc11-Cre), the control mice received the same dosage of tamoxifen. This is now clearly stated in the Methods section.

In each experiment the number of animals used (n) is missing. Please state in each figure how many mice are used or how often the cell culture experiments are repeated.

The number of mice and repetitions of experiments are now provided in each figure legends.

Figure 1: E/F Quantification of F4/80 stained macrophages in the immunohistochemisty and western blot analysis is missing.

Quantifications have been added as requested.

Moreover, to support the conclusion that these macrophages are M2 macrophages marker staining for M1 (CD11c/CD80) and M2 (CD206) should be performed. At least a VEGF staining should be performed.

The presence of M1 vs. M2 macrophages is an interesting question. Due to the short time allocated by the Editor to the revision(10 days), we couldn’t perform any additional staining. It would take 4-6 months to generate enough new mice for m1 vs. M2 quantification. We quantified VEGF on tissue lysate by ELISA in a small number of mice. VEGF level in ERK1 KO mice tissues collected 3 days after surgery had a slightly and not significant increase in VEGF than WT tissues (see below). Since the difference where not significant, we didn’t include the data in the manuscript.

Figure 3: E: Quantification of western blot analysis is missing. Immunohistochemistry staining for F4/80 should be performed to allow comparison to global Erk1 knockout mice.

Quantification was added. Due to the short time to do the revisions (10 days) we couldn’t perform any additional staining.

Furthermore, eNOS expression in endothelial cells should be analysed to compare to EC-specific erk2 knockout effect.

eNOS expression in ERK2 iECKO endothelial cells is provided on Fig. 4F.

Figure 4: Immunohistochemistry staining for F4/80 and VEGF should be performed to allow comparison to global Erk1 and EC/macrophage-specific Erk1-/- knockout mice.

Due to the short time to do the revisions (10 days) we couldn’t perform these additional experiments.

Figure 5: If pulmonary ECs from the different knockout mice are available, I would suggest analysing eNOS expression in these cells in addition to LSECs.

Unfortunately, we do not have pulmonary Ecs from different KO strains available.

Minor comments:

Introduction:

p1, line 26 ”is particular is” p1

The typo was corrected.

line 26-31, citation?

We added 2 citations:

Lee, C.W., et al., Temporal patterns of gene expression after acute hindlimb ischemia in mice: insights into the genomic program for collateral vessel development. J Am Coll Cardiol, 2004. 43(3): p. 474-82.

Scholz, D., et al., Contribution of arteriogenesis and angiogenesis to postocclusive hindlimb perfusion in mice. J Mol Cell Cardiol, 2002. 34(7): p. 775-87.

p1, line 45, ..signaling cascades VEGF.

The typo was corrected.

Methods:

Method description for isolation of aortic vSMCs and pulmonary ECs is missing. siRNA transfection protocol is missing.

The methods were added.

There is a typo on page 13, line 289, “beqads”

The typo was corrected.

Reviewer 2 Report

In this study Ricard and colleagues investigate the specific role of the ERK isoforms in arteriogenesis. They demonstrate that ERK1 is involved in macrophage infiltration following an ischemic event and leads to excessive but inoperative arteriogenesis. They also characterized the cell types responsible for this by using cell specific KO ERK1 mice and showed that loss of ERK1 in endothelial cells and macrophages contributes to increased macrophage infiltration and increased and dysfunctional arteriogenesis. They also characterized the role of ERK2 and their results suggest that endothelial cell deletion of ERK2 results in decreased arteriogenesis due to decreased endothelial cell proliferation due to decreased eNOS expression. This is a well conducted study there are however a few issues that need further clarification.

Major comments

The authors should provide quantification of the western blot protein analysis shown in Fig. 1F and 3E and statistical analysis of the results.

When characterizing the cell specific effects of Erk1 the authors demonstrate that, although deletion effect of Erk1 in ECs, SMCs and MΦs has not significant effect on arteriogenesis, deletion of Erk1 in ECs and MΦs affects arteriogenesis by inducing infiltration of MΦs resulting thus in not functional arteriogenesis. However, the effects on blood vessel diameter and F4/80 expression are smaller when compared to the results from the total Erk1 KO. Can the authors explain why?

Considering that the effects of the ERk1 deletion in ECs and MΦs on vessel diameter  are smaller compared to the total KO, is it possible that other cell types like SMCs are involved in the phenotype seen in the total Erk1 KO? Have the authors characterized the effects of Erk1 deletion in SMCs and MΦs simultaneously, or Erk1 deletion in ECs and SMCs simultaneously?

How total Erk1 deletion or specific deletion of Erk1 in ECs and MΦs result in increased MΦ infiltration and defective arteriogenesis? Is this due to changes in NFkb or Notch signaling? Is there an effect on MCP1 expression?

In figure 5A although quantification of the results suggest that Erk1 total deletion in Erk2(iEC-/-) does not have an additional effect on EC proliferation. Nevertheless, from the pictures it looks that in ECs with Erk1(total-/-) and Erk2(iEC-/-) deletion there are less proliferating ECs. Can the authors explain these discrepancies?

The authors characterized the effects of Erk2(iEC-/-) or Erk1(total-/-) in EC proliferation in response to FGF. What are the effects on ECs with Erk1(total-/-)+Erk2(iEC-/-) deletion? What are the effects on VEGF-induced EC proliferation?

What are the effects on migration of ECs treated with shRNAErk1?

Minor comments

In the figures 1D, 3D and 4E please correct ‘diamater’ it should be diameter

In all figures please indicate the number of mice or experiments analyzed.

In figure 5A although please indicate what do the different stainings indicate (green isolectin B4 and red BrdU?).

Author Response

Comments and Suggestions for Authors:

In this study Ricard and colleagues investigate the specific role of the ERK isoforms in arteriogenesis. They demonstrate that ERK1 is involved in macrophage infiltration following an ischemic event and leads to excessive but inoperative arteriogenesis. They also characterized the cell types responsible for this by using cell specific KO ERK1 mice and showed that loss of ERK1 in endothelial cells and macrophages contributes to increased macrophage infiltration and increased and dysfunctional arteriogenesis. They also characterized the role of ERK2 and their results suggest that endothelial cell deletion of ERK2 results in decreased arteriogenesis due to decreased endothelial cell proliferation due to decreased eNOS expression. This is a well conducted study there are however a few issues that need further clarification.

 We would like to thank the reviewer for its nice comments.

Major comments:

The authors should provide quantification of the western blot protein analysis shown in Fig. 1F and 3E and statistical analysis of the results.

Quantifications are now provided in all appropriate figure as requested.

When characterizing the cell specific effects of Erk1 the authors demonstrate that, although deletion effect of Erk1 in ECs, SMCs and MΦs has not significant effect on arteriogenesis, deletion of Erk1 in ECs and MΦs affects arteriogenesis by inducing infiltration of MΦs resulting thus in not functional arteriogenesis. However, the effects on blood vessel diameter and F4/80 expression are smaller when compared to the results from the total Erk1 KO. Can the authors explain why? Considering that the effects of the ERk1 deletion in ECs and MΦs on vessel diameter are smaller compared to the total KO, is it possible that other cell types like SMCs are involved in the phenotype seen in the total Erk1 KO? Have the authors characterized the effects of Erk1 deletion in SMCs and MΦs simultaneously, or Erk1 deletion in ECs and SMCs simultaneously?

We agree that the phenotype of ERK1 iEC Mɸ is less severe than the one in ERK1 KO. We cannot exclude that other cell types are involved. We added this point in the discussion section. It’s not clear what cell types one would go after to do additional experiments. It may well be leukocytes, lymphocytes or tissue-resident macrophages. Given already extensive number of KO lines in this study, the impossibility of identifying a correct target to go after and a relatively small size of the effect we would be looking for, we elected not to carry out additional studies. There is also a practical consideration of a 10-day revision deadline established by the journal. The work reviewer si asking for could easily take ~6-8 months.

How total Erk1 deletion or specific deletion of Erk1 in ECs and MΦs result in increased MΦ infiltration and defective arteriogenesis? Is this due to changes in NFkb or Notch signaling? Is there an effect on MCP1 expression?

We studied the notch and NFkB signaling in isolated endothelial cells from ERK1 KO mice and we haven’t seen any defect (not shown). We quantified MCP1 expression by ELISA in a small number of tissue lysates from ERK1 KO mice 3 days after surgery and found a statistically not significant increased expression compared to WT (see below). Since the difference was not significant, we didn’t include the data in the manuscript.

In figure 5A although quantification of the results suggests that Erk1 total deletion in Erk2(iEC-/-) does not have an additional effect on EC proliferation. Nevertheless, from the pictures it looks that in ECs with Erk1(total-/-) and Erk2(iEC-/-) deletion there are less proliferating ECs. Can the authors explain these discrepancies?

 We believe that direct measurement of proliferation is far more accurate that visual assessment based on a single panel in a figure. This is why we carried out proliferation quantification studies in vitro.

The authors characterized the effects of Erk2(iEC-/-) or Erk1(total-/-) in EC proliferation in response to FGF. What are the effects on ECs with Erk1(total-/-)+Erk2(iEC-/-) deletion? What are the effects on VEGF-induced EC proliferation?

As KO of Erk2 alone totally inhibits EC proliferation in response to FGF2, it does seem reasonable to carry out double Erk1/Erk2 KOs.
VEGF stimulation gave similar results to that of FGF2 stimulation (see below) but with less difference between stimulated and not stimulated cells. This is not surprising as VEGF is not a strong mitogen. We didn’t add the VEGF data in the manuscript as the conclusion is similar to the one with FGF2.

What are the effects on migration of ECs treated with shRNAErk1?

As shown on Fig. 4D, there is no migration defect on EC treated with shRNA against ERK1 alone or ERK2 alone. Migration in only affected when both ERK isoforms are knock down.

Minor comments

 In the figures 1D, 3D and 4E please correct ‘diamater’ it should be diameter

The typo was corrected.

In all figures please indicate the number of mice or experiments analyzed.

The number of mice and repetitions of experiments are now provided in each figure legends. 

In figure 5A although please indicate what do the different stainings indicate (green isolectin B4 and red BrdU?).

We added the requested information.

Round 2

Reviewer 1 Report

The authors sufficiently adressed the questions.